# Genome-Wide Analysis of the *Rab* Gene Family in *Melilotus albus* Reveals Their Role in Salt Tolerance

**DOI:** 10.3390/ijms24010126

**Published:** 2022-12-21

**Authors:** Caibin Zhang, Fan Wu, Qi Yan, Zhen Duan, Shengsheng Wang, Bao Ao, Yangyang Han, Jiyu Zhang

**Affiliations:** State Key Laboratory of Herbage Improvement and Grassland Agro-Ecosystems, Key Laboratory of Grassland Livestock Industry Innovation, Ministry of Agriculture and Rural Affairs, College of Pastoral Agriculture Science and Technology, Lanzhou University, Lanzhou 730020, China

**Keywords:** *Rab* genes, salt tolerance, heterologous expression, expression pattern, *Melilotus albus*

## Abstract

*Melilotus albus* is a high-quality forage, due to its high protein content, and aboveground biomass and salt tolerance. Rab (Ras-related protein in the brain) proteins are the largest GTPase family which play a key role in intracellular membrane transport, and many *Rab* genes have been identified in eukaryotes. The growth and distribution of *M. albus* are severely hampered by soil salinization. However, little is known about candidate genes for salt tolerance in *M. albus*. In this study, 27 *Rab* family genes were identified for the first time from *M. albus*, and divided into eight groups (Groups A-H). The number of introns in *MaRabs* ranged from one to seven, with most genes containing one intron. In addition, most MaRab proteins showed similarities in motif composition. Phylogenetic analysis and structural-domain comparison indicated that *Rab* family genes were highly conserved in *M. albus*. Members of the MaRab gene family were distributed across all eight chromosomes, with the largest distribution on chromosome 1. Prediction of the protein interaction network showed that 24 Rab proteins exhibited protein–protein interactions. Analysis of the promoter *cis*-acting elements showed that *MaRab-*gene family members are extensively involved in abiotic stress responses. RNA-seq data analysis of the *MaRab-*gene-expression patterns suggested that the *Rab* gene family possesses differentially expressed members in five organs and under salt stress, drought stress, and ABA (Abscisic Acid) treatment. Differentially expressed genes under drought stress, salt stress and ABA stress were validated by quantitative real-time PCR. Furthermore, heterologous expression in yeast was used to characterize the functions of *MaRab1* and *MaRab17*, which were upregulated in reaction to salt stress. In summary, this study provided valuable information for further research into the molecular mechanism of the response of *M. albus* to saline stress, as well as the possibility of developing cultivars with high salt-resistance characteristics.

## 1. Introduction

The agriculture and animal-product industry plays a very significant role in the world, and is increasingly developing as a modern area of economic activity [1]. However, forage plants often suffer from abiotic stress as a primary challenge. Soil salinization is a major problem in global agricultural production [2]. Usually, abnormal climate conditions, such as extreme drought or excessive application of chemical fertilizers, allow a high accumulation of salts on the soil surface, interfering with the ability of plant roots to function properly in nutrient uptake and thus undermining the ability of plants to accumulate or even maintain their current biomass, ultimately causing imbalanced agroecosystems [3]. There is a direct link between salinity stress and dysregulation of the plant cell-cycle, and it leads to a low proliferation rate of cells [4]. Salt stress is a fatal challenge for forage-plant production, and reduces profit in global agriculture year after year. In response to saline stress, plants could develop several strategies to survive and maintain biomass; for example, by regulating the expression of genes that control levels of cytoplasmic salt-ion concentration [5].

*M. albus* is an annual or biennial herb of the Meadowsweet genus (*Melilotus*) in the family Fabaceae [6]. It can be used as green fertilizer and as forage. *M. albus* has a well-developed root system that penetrates deep into the soil and grows well in clay soil, sandy soil, and other infertile soil [7]. *M. albus* is widely distributed, and has strong stress-resistance [8,9,10]. *M. albus* has strong vitality and adaptability, with excellent characteristics, such as drought tolerance, cold tolerance, infertile-soil tolerance, and salinity tolerance [11,12]. As long as the salt content does not exceed 0.30%, *M. albus* can grow and develop very well [13].

Rab proteins, as a part of the *Ras* superfamily of small GTPases, have essential attribute characteristics similar to those of small GTPases in other subfamilies [14]. The Rab protein is a small GTP-binding protein composed of approximately 200 amino acids [15]. Rab proteins are present in all eukaryotes. Sequence analyses of *Rab* genes in different species have indicated that the Rab protein was highly conserved in eukaryotic evolution, although it shows quantitative diversity and functional differentiation among different organisms [16]. Rab proteins are very conserved throughout species, characterized by a conserved G-domain and highly varied N- and C-terminal sections [17]. Rab proteins interact with upstream regulators and specific downstream-effectors, and are coupled with GTP binding and hydrolysis processes acting at different stages of vesicle transport [18].

The plant endomembrane system does not only participate in the regulation of cell walls, plasma membranes, and vesicle biosynthesis, but also plays a crucial role in multiple stress responses [19]. *McRab5b*, of the *Rab5* family in *Mesembryanthemum crystallinum,* has been induced in response to 400 mmol/L NaCl treatment [20]. Rice *OsRab7*, Arabidopsis *AtRab7*, wolfsbane *PgRab7*, and salt plant (*Aeluropus logophiles*) *AlRab7* have been induced to be expressed in reaction to cold, salt, drought, and ABA (Abscisic Acid) treatments [21,22,23,24]. Thus, the induced expression of *Rab7* in different stress-environments indicates that this gene is involved in adaptation to stresses. When the four major Arabidopsis *RabA1* members were knocked out together, these *RabA1B*-dominant inactivating mutants exhibited a hypersensitive response to salt [25]. Furthermore, experimental results show that RabA1 proteins are closely related to salt-stress responses of plants, by mediating the transport of substances between the plasma membrane and the trans-Golgi. [26]. Overexpression of the *AtRab7* gene in Arabidopsis increased plant tolerance to salt stress, and the buildup of reactive oxygen species was minimized under salt stress. Transgenic plants with Arabidopsis *AtRab7* overexpression exhibit higher aboveground sodium levels and Na^+^ accumulation in their vesicles, thereby maintaining low cytoplasmic toxicity and increasing plant tolerance to salinity stress [22]. Moreover, overexpression of the constitutive activation-mutant *PtRabE1b*(Q74L) confers salt tolerance in poplar [27].

In this study, we analyzed the evolutionary relationships, structural features, and expression patterns of *MaRab*, using bioinformatics and transcriptomic data concerning Rab proteins in *M. albus*. The functions of the *MaRab1* and *MaRab17* genes in the development and stress response of *M. albus* were verified using the validation of heterologous expression in yeast.

## 2. Results

### 2.1. Identification and Sequence Analysis of Rab Genes in M. albus

We initially obtained 29 candidate *MaRab* genes. After filtering, in total, 27 *MaRab* family members were discovered in the *M. albus* genome (Appendix A). *MaRab* genes coded for proteins with molecular weights of 202 aa (22.43 kDa) to 235 aa (25.87 kDa) (Appendix A). Their hydrophilicity ranged from −0.441 to −0.145, while isoelectric points (pIs) of MaRab proteins ranged from 4.85 to 6.84. Twenty-two MaRab proteins (81%) were most likely found in the cytoplasm, according to the anticipated subcellular localizations based on the presence of a signaling domain in the amino acid sequence of the MaRab proteins, while 2, 2 and 1 Rab proteins were most likely found in the chloroplast, extracellular space, and Golgi apparatus, respectively.

The predicted chromosome localization of *MaRab* genes was assigned differentially to the eight chromosomes. Seven *MaRab* genes were mapped to Chr (Chromosome) 1 (Figure 1), and *MaRab27* was the only one that was mapped to Chr 7. The 27 MaRab proteins associated with the genes were given names according to the predicted positions of all proteins on Chr 1–8 (from top to bottom).

The phylogenetic tree indicated that all *Rab* gene members were classified into eight phylogenetic groups. The 27 *Rabs* from *M. albus* were unevenly clustered into eight groups. There were twelve, four, two, two, two, two, two and one members in Groups A, G, B, C, D, F, H and E, respectively.

To study the evolution of the *MaRab* gene family, we constructed a phylogenetic tree with the Rab proteins of *M. albus, M. truncatula, O. sativa, A. thaliana, G. max, C. arietinum* and *L. corniculatus*. All Rab proteins were divided into eight groups, A-H (Figure 2). The genetic evolutionary-distances between these genes are listed in Appendix A. Evolutionary analysis of *Rab* genes from seven plant species revealed that Arabidopsis was the species with the highest number of members in group A (25), followed by *O. sativa* and *G. max* (14 members each), while *M. albus*, *C. arietinum* and *L. corniculatus* had 12 members and *M. truncatula* was the species with the fewest members in group A (10). Seven species had a similar number of *Rab* members in Group B, C, D, E and F. In contrast, *A. thaliana* possessed a higher number of *Rab* members in Group G and Group H than was the case with the other six species. This result seems to suggest that the numbers of members in Group A, Group G and Group H shrunk to a greater extent for Arabidopsis than for the other six species.

### 2.2. MaRab-Protein-Sequences Analysis

Potential motifs were predicted, to better understand the structural properties of *MaRab*, and 15 different motifs were found (Figure 3A). The complete PSPG-box motif was present in motif 1, which was shared by all members. Motif 15 was only found in Group A, which had only two members; motif 6 was present in 25 members; motif 8 was present in 26 members; and motifs 1, 2, 3, 4, and 5 were noticeable in all members. Phylogenetic research revealed that groups have a similar conserved motif, as shown in Appendix A. These particular patterns could, in part, result in distinct functional properties of the *Rab* genes in *M. albus*. Additionally, the structure of 27 *MaRab* gene was examined in terms of intron and exon organization (Figure 3B). According to the research, *MaRab* genes had two to eight exons, and various numbers of introns (Appendix A). The closely related clustering of *MaRab* genes in the same group showed close resemblance in intron number and exon length, which was analogous to the phylogenetic relationship in Figure 1. With twelve members, Group A of the *Rab* groups had the most genes with intron deletion, followed by Group G, with four individuals. Twelve of the *Rabs* in Group A had one intron in total.

To clarify the *MaRa-* gene transcriptional regulation, the promoter region of 27 *MaRabs* was analyzed for *cis*-acting elements (Figure 4). In total, 401 *cis*-acting elements were found in this analysis, and could be grouped into 24 sets according to their described roles in the bibliography (Appendix A). Among all the *cis*-acting elements of the promoters in *MaRab* genes, the three sets with the highest numbers of elements were associated with MeJA (Methyl Jasmonate)-responsiveness (80, 20%), light responsiveness (55, 13.7%) and anaerobic induction (54, 13.7%). On the other hand, the promoter of the *MaRab7* gene contained the largest number of *cis*-acting elements (34). These findings suggested that *MaRab* gene members in *M. albus* play a crucial role in hormone regulation, and may be implicated in multiple stress-responses. Among all groups of the *MaRab* gene family, the promoters of Group A genes contained the highest number of *cis*-acting elements, at 172 (42.9%) (Appendix A). Promoters of the Group A gene also contained the highest number and percentage of *cis*-acting elements associated with MeJA responsiveness (36, 45%), anaerobic induction (34, 63%), light responsiveness (16, 29.1%), ABA responsiveness (15, 34.1%), salicylic acid responsiveness (10, 58.8%), and low-temperature responsiveness (10, 43.5%), although Group H contained only one *cis*-acting element for flavonoid biosynthetic gene-regulation. From the prediction of potential *cis*-acting elements, we also found that Group H contained a gibberellin-responsiveness *cis*-acting element. In addition, there were 51 and 48 *cis*-acting elements of the Group C genes and Group D genes, respectively. In the promoters of Group C genes, there were seven, nine and ten *cis*-acting elements related to anaerobic induction, light responsiveness and MeJA-responsiveness. In the promoters of Group D genes, there were ten, ten and sixteen *cis*-acting elements which were associated with ABA responsiveness, light responsiveness and MeJA-responsiveness. In the promoters of Group F genes, there were eleven, eight and six *cis*-acting elements related to light responsiveness, ABA responsiveness and MeJA-responsiveness. The promoters of Group E genes had the lowest number of *cis*-acting elements with only six. However, *cis*-acting elements contained in the promoters of the other groups were not centrally associated with any of the stresses.

We also processed the protein 3D-structure prediction for the Rab proteins from *M. albus* (Appendix A). The comparison revealed that, for the most part, *MaRab-*gene-family members shared similar 3D structures, and the similarity was even higher among gene family members within the same group. For example, *MaRab11*, *MaRab 22*, *MaRab 24* and *MaRab 27*, which belonged to Group G, had very high similarity of 3D structure among the four gene-family members. In addition, we also found a large number of protein interaction cues in the *MaRab* gene family, by constructing a protein interaction network (Appendix A). As observed, all 24 members of the *MaRab* family, except *MaRab11*, *MaRab14* and *MaRab15*, had mutual interactions, with *MaRab21* and *MaRab23* having the highest number of interactions, 23 pairs of interactions, while 45.8% of the other *MaRab* gene family members had 13 pairs of interactions, 33.3% had 22 pairs of interactions, and 12.5% had 21 pairs of interactions.

To clarify the sequence characteristics of the MaRab proteins, we performed multiple sequence alignments. We identified the Rab domain with five Rab-specific motifs, I–V, which were identical in at least 80% of the 27 Rab proteins (Figure 5). This finding suggests that the protein sequences of the *MaRab-*gene family members identified in this study are highly conserved at certain amino acid sites.

### 2.3. Evolutionary Analysis of the MaRab Gene Family

To clarify the evolutionary progression of *MaRab-*gene family members, we performed synteny analysis. There were 13 syntenic pairs in *M. albus*, one of which was a homologous gene pair, *MaRab24/MaRab2* (Appendix A). Moreover, 12 syntenic pairs were nonhomologous gene pairs, and *MaRab25* participated in two gene pairs (Figure 6). To determine if Darwinian positive selection participated in the divergence of *MaRab* genes following duplication, the nonsynonymous to synonymous substitution-rate ratios (*Ka/Ks*) were determined. The *Ka/Ks* were much less than one, indicating that purifying selection is important in the duplication of *MaRab* gene pairs (Appendix A).

### 2.4. Expression-Pattern Analysis of MaRab Genes

Plant organs typically have varying amounts of gene expression, and diverse functions for various metabolites. RNA-seq data from *M. albus* showed different expression patterns for *Rabs* in each of the organs studied (Figure 7A). Except for eight genes, nineteen *MaRab* genes had expression values (FPKM) above one in at least one of the organs studied. The analysis of differentially expressed genes revealed that some genes were highly expressed in some specific organs, such as six *MaRab* genes that were preferentially expressed in roots (Appendix A). Each *Rab* gene displayed a distinct expression-pattern, which was linked to the evolutionary groups.

The *MaRab* gene family was found to respond to salt, drought and ABA stresses, based on the number of all the upregulated, and the expression of *MaRab* genes was downregulated in roots and shoots when exposed to salt, drought, and ABA treatment. (Appendix A). As observed, a comparison of the number of upregulated DEGs (differential gene expressions) with the number of downregulated DEGs revealed greater upregulated DEGs than downregulated DEGs in both the shoot and root organs. Among those in the shoots, 22% of the upregulated *MaRab* genes were co-expressed under ABA, salt and drought stresses. Those specifically expressed under ABA, salt and drought stresses accounted for 15%, 22% and 22% of the upregulated *MaRab* genes, respectively. This finding indicates that *MaRa-*gene family members are extensively involved in abiotic-stress-response processes in *M. albus.* The heatmap shows the expression of *Rab* members matching the previous grouping at diverse time-points under salt, ABA, and drought stresses (Figure 7B).

*MaRab11*, *MaRab13*, *MaRab17, MaRab22* and *MaRab25* were significantly increased after 3 h of 250mM NaCl-irrigation in the shoots, and *MaRab1*, *MaRab5*, *MaRab6*, *MaRab12*, *MaRab19* and *MaRab23* were significantly downregulated after 3 h of NaCl treatment in the shoots. In addition, *MaRab1*, *MaRab5*, *MaRab6, MaRab11, MaRab17* and *MaRab22* were significantly increased after 3 h of NaCl treatment in the roots, and *MaRab12*, *MaRab13*, *MaRab19*, *MaRab23* and *MaRab25* were significantly downregulated after 3 h of NaCl treatment in the roots. These genes are assumed to be associated with salt stress. To explore the expressions of *MaRab* genes under salt stress, we studied the gene expression in the leaves of JiMa46 under NaCl stress, using qRT-PCR data. The qRT-PCR results showed comparable expression patterns to the RNA-seq data. The change trends in *MaRab* gene expression at different times were consistent (Figure 8).

### 2.5. MaRab1 and MaRab17 Improve Salt Tolerance in Yeast

An analysis of qRT-PCR after 250 mM NaCl-treatment for 3 h showed that the expression of *MaRab1* in shoot and root increased 3.6 times and 12.1 times, respectively, and the expression of *MaRab17* in shoot and root increased 2.4 times and 8.2 times, respectively, compared with CK. We expected that the proteins encoded by these two genes would be critical in the salt stress response of *M. albus*. The effects of *MaRab1* and *MaRab17* on yeast growth and stress resistance (NaCl) were investigated in yeast cells using pYES2−*MaRab1* and pYES2−*MaRab17* constructs. Under normal culture conditions, there was no divergence between the transformed lines and the empty vector lines. Treatments with 5 M NaCl had a significant impact on the development of transformed yeast, and we found that changed yeast cells demonstrated prominent resistance under 5 M NaCl treatments, particularly at 10^−4^ dilution (Figure 9).

## 3. Discussion

*Rabs* have important roles in expansion, maturation and abiotic-stress responsiveness [28]. They play a significant role in controlling interactions with vesicular traffic, and they are essential for secondary metabolite biosynthesis, collection, and transportation [29]. Even though *Rabs* have been discovered in some species, the classification and protein characterization of *Rab* gene members in *M. albus* has never been published [30,31]. Arabidopsis *Rab* was the primary species utilized to investigate evolution in an earlier study. Its *Rab* phylogenetic tree was classified into eight families (A-H) [32]. In our research, we discovered 27 *Rab* genes. A total of 27, 30, 27 and 28 *Rab* genes were discovered in *M. truncatula*, *G. max*, *C. arietinum* and *L. corniculatus*, respectively. According to an evolutionary study, 27 *Rab*s from *M. albus* could also be grouped into eight groups. There were a larger number of members in Group A than in the other groups. During plant development, Group A seems to have expanded more than the other groups. In a previous study, *A. thaliana* contained 25 *Rab* members of Group A, accounting for 45% of the total number of *AtRabs* [32]. Group A is the largest group of the *Rab* family. Thus, we speculated that a similar situation could occur in *M. albus*. Our research also revealed that Group A was a sizable group, comprising 12 members of the *Rab* family, and accounting for 44% of the Rab proteins in *M. albus*. In a previous study, 67 *Rab* members were identified in poplar [27]. The number of Rab proteins in the five plants identified in this study, including *M. albus*, was smaller than the numbers in Arabidopsis and *Populus*. This indicated that the *Rab* genes did not expand in *M. albus*.

Intron gain- and loss-events, along with intron locations and phases relative to protein sequences, provide essential insights into evolution [33]. In a previous study, the intron numbers of most *Rab* genes from the same group in *Gossypium* were nearly identical, with only several genes representing exceptions [34]. In our study, intron mapping of the 27 *MaRabs* revealed that the number of introns was significantly different among *MaRab* genes. A total of 12 (44%) of the *Rabs* in Group A contained one intron, indicating that the bulk of preserved introns were ancestral elements with stable phases, as deletion and insertion of tiny DNA sequences that produce a phase transition might lead to alterations in gene expression, and thus be deleted by biological evolution [35]. In a previous study, some *cis*-acting elements that played a crucial role in plant growth and maturation were identified during the promoter analysis of genes. TCP promoters, for example, have been revealed to contain *cis*-acting elements implicated in hormone response, signal transduction, and light response, as well as growth- and maturation-regulation and metabolism regulation [36]. In our study, the promoter-regions analysis revealed that some of the *Rab* promoters contained stress-related elements, including cell-cycle regulation, defense, auxin, gibberellin, light, low-temperature, salicylic acid, abscisic acid responsiveness, and MeJA-responsive *cis*-acting elements. Protein-interaction-network prediction is important for studying gene regulation in response to various stresses, as well as growth and maturation in species [37]. We also performed protein-interaction prediction and found 24 protein interactions between members of the *MaRab* gene family. This prediction was based on information from curated databases, experimentally determined findings, gene neighborhoods, gene co-occurrence, text mining, coexpression and protein homology. These interaction relationships may facilitate progress in the study of the *Rab-*gene-family-related regulatory networks.

The patterns of gene expression can also provide information for screening target genes, and can help predict gene function. Recent research discovered considerably distinct expression patterns in every category of the *Rab* family in cotton, suggesting that cotton *Rabs* may participate in the plant stress-responsiveness mechanism [34]. In our study, the analysis of *Rab* expression patterns in various organs revealed that the expression patterns of *MaRab* were linked with the phylogenetic groupings. Expression levels of many *MaRab* genes changed in reaction to stress treatment, and these genes were mostly from Group A. In the response to salt stress, *MaRab1*, *MaRab5*, *MaRab11*, *MaRab13* and *MaRab17* were significantly upregulated, and *MaRab19*, *MaRab23* and *MaRab25* were significantly downregulated by salt treatment, in the roots after 3 h. These *MaRab* genes are most likely linked to salinity stress.

Plants are exposed to numerous stresses throughout their lives, and have expanded their capacity to cope with adverse external conditions [38]. A previous study showed that several *Rab* family members participated in the reaction to environmental stresses [39]. For example, it has been observed that *RabA1* members are essential for resistance to salinity stress when performing salt-stress assays of transgenic plants, and *A. thaliana RabA1* quadruple mutants and organs from *A. thaliana* producing the dominant deletion mutant of *RabA1b* were sensitive to salt stress with 15 mM NaCl [26]. In this study, we discovered that the altered yeast cells were highly resistant to 5 M NaCl, especially under 10^−4^ dilution, suggesting that *MaRab1* and *MaRab17* play a significant role in the salt responsiveness. To further understand the function of *Rab* genes, we found that *Rab11a*, a homologue of *MaRab1* and *MaRab17*, primarily participated in cytosolic vesicular transport and endocytosis [40]. The Rab protein is a vital particle regulator inside the cell, regulating vesicle trafficking in a variety of ways to promote the formation as well as maintenance of apical polarity [41,42]. In addition, many plant *Rab-*gene members have been functionally identified, including interactions with vesicular transport. For example, members of *RabA1* regulate traffic around the plasma membrane and the trans-Golgi network [26], indicating that *Rab* contributes significantly to metabolite transport in reaction to sodium stress [20,43,44]. Therefore, we speculated that *MaRab1* and *MaRab17* can be involved in the cellular vesicular transport and endocytosis that are part of the regulatory activities of plants in response to salt stress.

## 4. Materials and Methods

### 4.1. Materials, Growth Conditions, Treatment, and Sampling

*M. albus* (line JiMa46) seeds were treated with sulfuric acid for 4 min to break the seed coat and sterilized in 5% NaClO solution for 10 min and then placed in a sterilized solid medium with half Murashige and Skoog (1/2 MS). They were placed in growth chambers with the environment of 16 h light photoperiod, 25 °C temperature, and 30% relative humidity, after 4 days of vernalization at 4 °C. After a 14-day germination period, the seedlings were transplanted to plastic pots (peat soil/vermiculite = 3:1) in a greenhouse for growth with the environment of 16 h light photoperiod and 25 °C temperature. We performed all of the experimental treatments 6 weeks after germination. Plants were irrigated with 250 mmol/L NaCl, to imitate salt stress. Samples were collected in three biological replicates from seedling leaves and roots at three time-intervals (0, 3, and 24 h). All samples were promptly placed in liquid nitrogen and later stored in an ultra-low temperature refrigerator at −80 °C.

### 4.2. Genome-Wide Identification of Rab Genes in M. albus

The Phytozome 13 database was used to obtain protein sequences of Rab proteins of Arabidopsis and *Oryza sativa* [45]. In addition, the NCBI database was used to obtain gene sequences and coding sequences (CDSs) [46]. To identify the candidate *MaRab* members, local BLASTN and BLASTP searches were conducted, using the CDS and protein sequences of Arabidopsis and *O. sativa* as queries, and an e-value cutoff of 1 × 10^−5^ was used for homologue recognition. Similarly, the *Rabs* of *Medicago truncatula, Glycine max, Cicer arietinum* and *Lotus corniculatus* were identified. Protein sequences were uploaded to the Pfam website to verify the candidate *Rab* genes with default parameters [47]. Rab candidate proteins that lack the Rab structural domains were dropped manually*. M. albus* protein sequences, gene sequences, and coding sequences (CDSs) were obtained from a previous study [48]. Local BLAST searches were carried out, with an e-value cutoff of 1e^−5^ utilized for *Rab*-homologous-sequence recognition. In this study, *Rabs* of Arabidopsis, *O. sativa*, *M. truncatula, G. max, C. arietinum* and *L. corniculatus* were used to construct a phylogenetic tree with *MaRabs*. The databases and websites mentioned above are listed in Appendix A. The evolutionary tree was constructed using MEGA software [49].

### 4.3. Chromosomal Location, Motif Analysis, Protein 3D-Structure Prediction, Gene Structure and Protein-Interaction Analysis of the Rab Genes in M. albus

Using TBtools and R with default parameters, each of the *MaRab* genes was marked on the genome of *M. albus* to indicate its position on the chromosome [48]. MEME was used to upload protein sequences of *Rab* family members, and was utilized to identify conserved motifs of *Rabs* with the default parameters [50]. WebLogo was utilized to create *Rab* sequence logos from the multiple alignment sequences [51]. The protein sequences of *MaRab-*gene family members were analyzed in this study using five Rab-specific conserved structural-domains discovered in previous *Rab* protein-related studies [52,53]. On the Swiss-Model website, 3D structures of Rab proteins were predicted [54]. In addition, the GeneStructure Display Server was used to compare CDSs with gene sequences of *Rab,* to establish and visualize all gene structures [55]. A protein-interaction network was generated by STRING for *MaRab* genes [56]. The databases and websites mentioned above are listed in Appendix A.

### 4.4. Characteristics of Rab Proteins, Analysis of Promoter Cis-Acting Elements and Synteny Analysis of Rab Genes

The Protsacle tool was utilized to predict the primary structure of Rab proteins [54]. To predict the subcellular localization of Rab proteins, the CELLO website was used [57]. The PlantCARE website and the 2kb sequence of the *MaRab* promoter were used to recognize the *cis*-acting elements [58]. BLAST (e-value < 1 × 10^−10^) was utilized to align genes within families, and the closest matching-gene match was considered as a homozygous pair. TBtools was utilized to calculate nonsynonymous substitution rates (*Ka*) and synonymous substitution rates (*Ks*) for the gene pairs [59]. MCScanX in TBtools with the default parameters was utilized to determine syntenic pairs of *M. albus* genes. Following that, TBtools was used to visualize the syntenic relationships. The databases and websites mentioned above are listed in Appendix A.

### 4.5. Transcriptomic Data Analysis and Quantitative Real-time PCR Analysis

Transcriptome data of *MaRab* genes in *M. albus* under ABA, drought and salt stresses and in different organs (roots, leaves, seeds, flowers and stems) were obtained from our previous studies [6,60]. The accession numbers of the transcripts corresponding to the 27 *MaRab* genes in these transcriptomic datasets are consistent with the gene ID numbers in Appendix A. The heatmap of *MaRab* gene expression in different organs and treatments was obtained from TBtools [59]. Total RNA was extracted from the roots and shoots of salt-treated, ABA-treated, drought-treated and control seedlings, by applying TransZol reagent (TransGen Biotech, Beijing, China). Using a TIANScript II RT Kit, cDNA synthesis and reverse transcription were conducted on 1 μg of the amount of total RNA (Tiangen, Beijing, China). Hieff^®^ qPCR SYBR^®^ Green Master Mix (No Rox) was used to perform qRT-PCR assays on each cDNA template (Yeasen Biotech Co., Ltd., Shanghai, China). The 2^−ΔΔCT^ method was used to determine the calculation of expression levels for the control group [61]. In this experiment, the *tubulin* gene was utilized as an internal control. The primers listed in Appendix A were used for this experiment. Three technical replicates were utilized for each biological triplicate. The significance analysis of relative expression levels was conducted by using the ANOVA function in IBM SPSS Statistics software (version 20). Multiple tests were performed using the Duncan method through variance analysis of different comparison groups.

### 4.6. Validation of Heterologous Expression in Yeast

First, we amplified all the coding sequences of *MaRab1* and *MaRab17* from JiMa46 to generate pYES2−*MaRab1* and pYES2−*MaRab17* constructs. The recombinant pYES2−*MaRab1* and pYES2−*MaRab17* plasmids produced using ClonExpress^®^ MultiS One Step Cloning Kit (Vazyme Biotech Co., Ltd., Nanjing) as well as the empty pYES2 plasmid were then transformed into *Saccharomyces cerevisiae* strain INVSc1, after sequence confirmation. After being grown separately for 36 h at 30 °C, both yeast cultures were collected separately in synthetic complete (SC)−Ura liquid medium containing 2% (m/v) galactose. Next, the yeast collection was incubated with SC−Ura containing 2% galactose and adjusted to A_600_ = 1 for stress analysis. Subsequently, the cells were resuspended in 5 M NaCl h at 30 °C for 36. At the same time, equal amounts of yeast cells were resuspended in 200 μL of sterile water at 30 °C for 36 h, to serve as a control. The cells were subsequently incubated in serial dilutions (10^0^, 10^−1^, 10^−2^, 10^−3^, 10^−4^, 10^−5^, 10^−6^) and finally spotted on SC-Ura agar plates [62]. Then, the yeast was allowed to grow for 2−3 days at 30 °C on this plate. Yeast solutions treated and diluted to 1:10 were grown on SC−U/2% (*w*/*v*) glucose agar plates at 1 cm intervals. After 3 days, pictures were taken to check colony formation and growth.

## 5. Conclusions

In total, 27 *MaRab* genes were discovered in *M. albus*. The phylogenetic, intron–exon and motif analysis revealed discrepancies between *MaRab* family groups. *MaRab* members were exposed to purifying selection, as well as several that were functionally redundant. Studies of the *MaRab* promoter region found that it may be involved in stress, maturation, and hormone responsiveness. The levels of gene expression in various organs were measured, and some *MaRab* members showed organ-specific expression. Furthermore, we expected that *MaRab1* and *MaRab17* would be involved in the regulation of salt stress via vesicular transport, based on the validation of heterologous expression in yeast. These findings offer a helpful framework for comprehending the development, biological function, and possible biological function of the *MaRab* gene family, under salt stress.

## Figures and Tables

**Figure 1 ijms-24-00126-f001:**
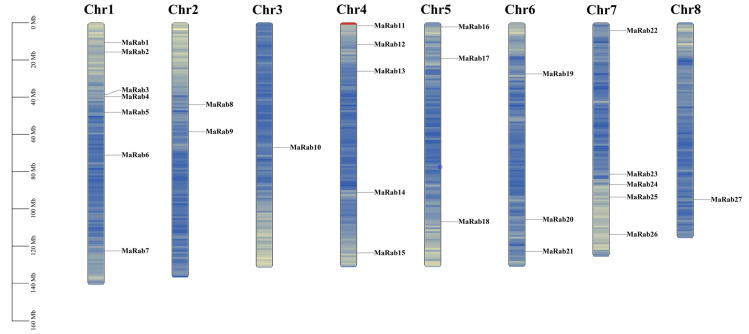
The chromosomal location of the *MaRab* genes.

**Figure 2 ijms-24-00126-f002:**
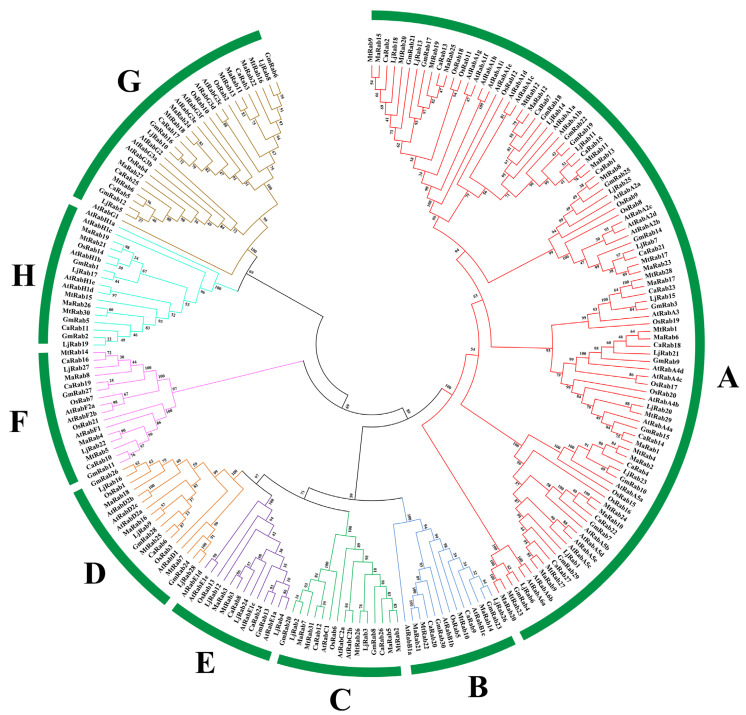
Phylogenetic analysis of *Rab* genes among *M. albus*, *A. thaliana*, *M. truncatula*, *O. sativa*, *G. max*, *C. arietinum* and *L. corniculatus*. The phylogenetic tree was constructed using the MEGA7 program with 1000 bootstrap replicates. The clusters of different colors in the figure represent different groups. These members were clustered into 8 groups (A–H). The number on the branch indicates the degree of reliability of this branch.

**Figure 3 ijms-24-00126-f003:**
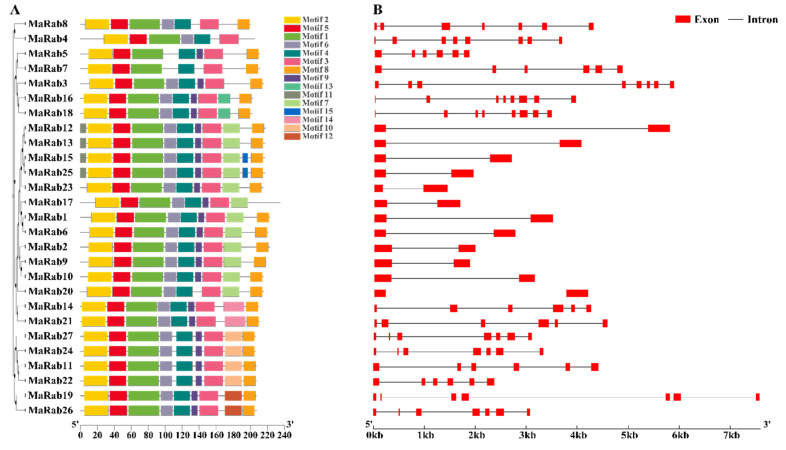
Distribution of conserved motifs and intron–exon structure of *MaRab* genes in *M. albus*: (**A**) Illustration of the conserved structural-domains in MaRab proteins. Different colors represent different kinds of conserved structural-domains. The nonconserved sequences are represented by the black lines. It displays 15 conserved domains and their associated motifs in the Rab protein sequences. (**B**) *MaRab* gene exon–intron structures. A black line designates the intron, while red boxes designate the CDS exon. At the bottom, a length scale for genes and proteins is displayed.

**Figure 4 ijms-24-00126-f004:**
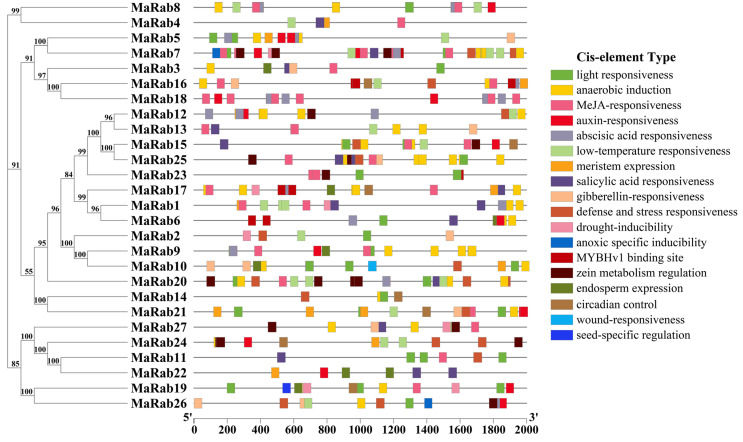
The *cis*-acting elements (−2000 bp) of 27 *MaRab* genes in *M. albus*. Different colored boxes represent different types of *cis*-acting element types. A scale of promoter length is shown at the bottom. The number on the branch indicates the degree of reliability of this branch.

**Figure 5 ijms-24-00126-f005:**
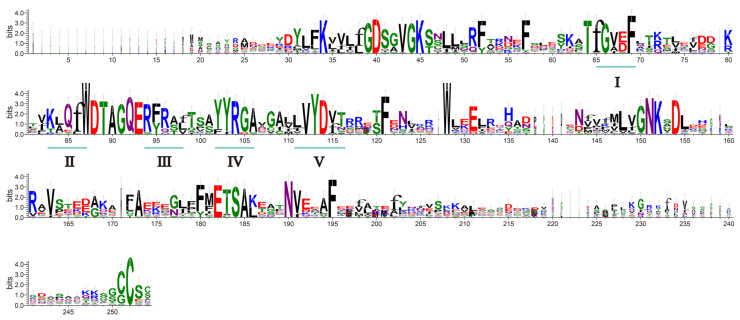
Web logo representing the PSPG-box motif of *MaRabs*. The five lines (**I**–**V**) represent conserved PSPG-box motifs shared by the Rab proteins. The logo is mainly composed of symbol stacks. Each location in the sequence has a stack. The overall stack height indicates the sequence’s conservation at that place, whereas the symbol height inside the stack indicates the relative abundance of each amino acid at that location.

**Figure 6 ijms-24-00126-f006:**
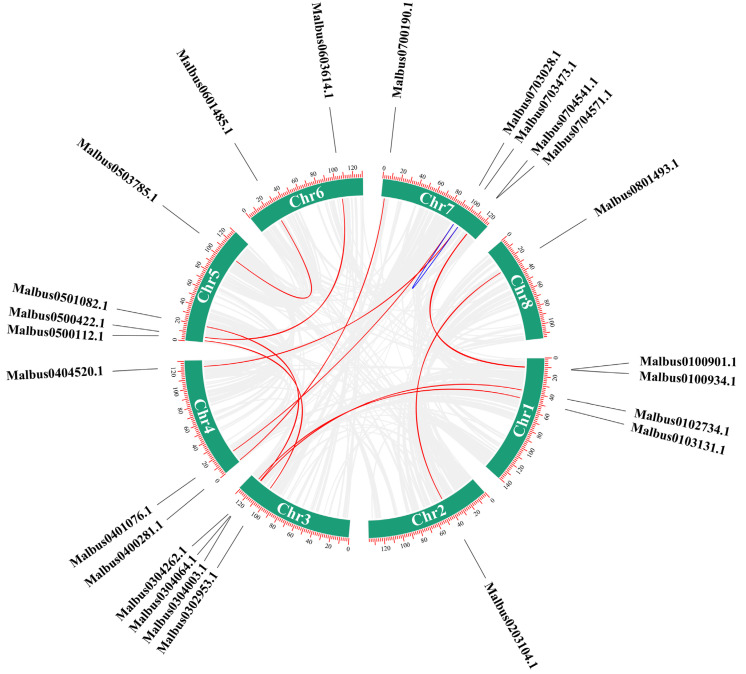
Synteny analysis of the *MaRab* genes from *M. albus*. The red lines indicate gene pairs between different chromosomes, while the blue lines are on the same chromosome. In the figure, the eight *M. albus* chromosomes are represented by closed circles in green. The numbers and scales on the chromosomes represent the position of the genes on the chromosomes. The ID numbers of the genes are located at the periphery of the chromosomes.

**Figure 7 ijms-24-00126-f007:**
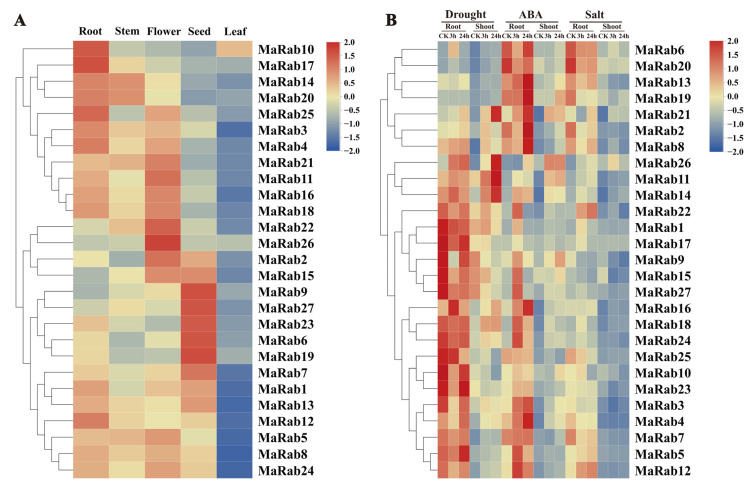
Expression pattern of *MaRab* genes: (**A**) patterns of *MaRab* gene expression in root, leaf, flower, stem and seed. Data were extracted from transcriptome datasets and clustered using TBtools. (**B**) Response of 27 *MaRab* genes to salt, drought and ABA treatments. Data were extracted from transcriptome datasets and clustered using TBtools. The heatmap depicts the relative levels of *MaRab* transcripts during salt, drought and ABA stress. Normalized expression values (FPKM) were obtained.

**Figure 8 ijms-24-00126-f008:**
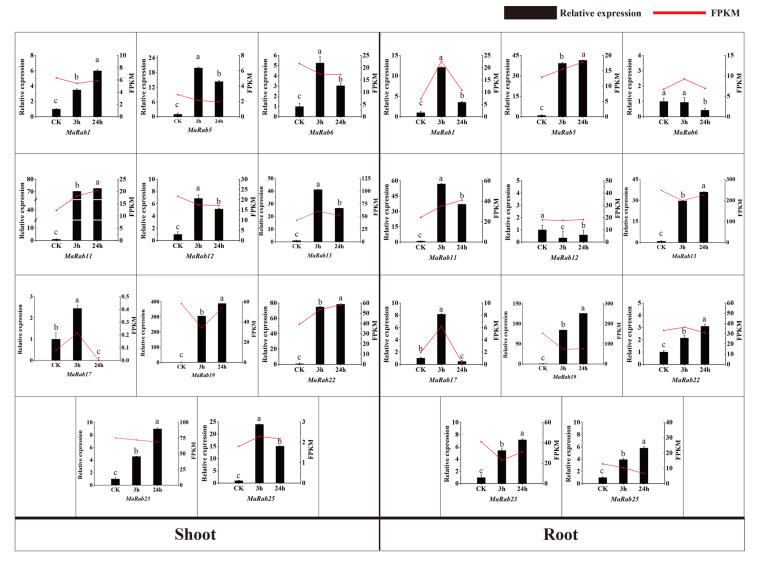
Gene expression of eight *MaRabs* in shoots and roots under NaCl treatments using qRT−PCR. Different small letters above the columns indicate significant (*p* ≤ 0.05) differences between various treatments. The control is represented by CK. The red lines represent the expression values (FPKM) from RNAseq data, and the values given are the mean ± standard deviations of three replicates.

**Figure 9 ijms-24-00126-f009:**
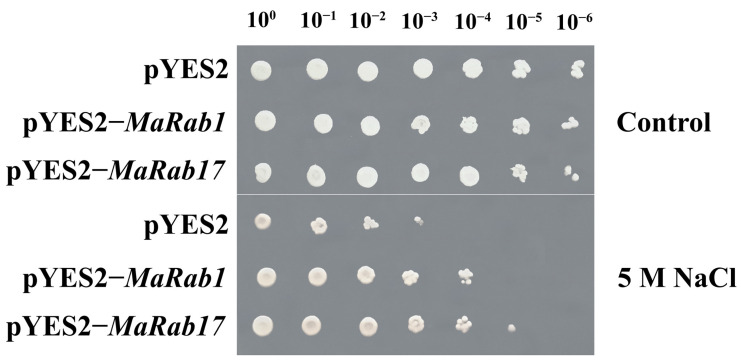
Analysis of NaCl treatment of *MaRab1* and *MaRab17* genes in yeast heterologous expression compared to the blank pYES2 line. After 36 h of resuspension under control and 5 M NaCl, consecutive dilutions (10^0^, 10^−1^, 10^−2^, 10^−3^, 10^−4^, 10^−5^, 10^−6^) of yeast were spotted onto SC−Ura solid medium. Sterile water was used as the control.

## Data Availability

The data presented in this study are openly available: *M. albus* genome data are available from NCBI, with BioProject ID: PRJNA674670. Transcriptome sequences of tissues have been submitted to the NCBI SRA under bioproject PRJNA647665.

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
