# Peer review of "Genome-Wide Analysis of the Rab Gene Family in Melilotus albus Reveals Their Role in Salt Tolerance"

_ijms, 2022, doi:10.3390/ijms24010126_

Round 1

Reviewer 1 Report

Manuscript entitled “Genome-Wide Analysis of the Rab Gene Family in Melilotus albus Reveals Their Role in Salt Tolerance” by Zhang et al describes the identification and characterization of homologues of  Rab proteins (GTPases with roles in in intracellular membrane transport,)  in the genome of the forage plant Melilotus albus. The authors performed this characterization first with several in silico analysis, including chromosomal location, Phylogenetic analysis, conserved motif analysis, promoter analysis, and evolutionary. This analysis was complemented by expression analyses in silico using transcriptome data and describe their expression profiles in tissues and abiotic stress conditions. Also, the expression profiles of some MaRab genes were also checked by qRT-PCR. Finally two of these genes were also used in yeast expression assays. Regarding the in silico analysis and expression studies, the proposed approaches seen both correct for the most part. The results obtained are interesting, and they can be an starting point in the obtention of new data that could help to unravel which role could take part the Rab proteins in the stress response in plants. However, in my opinion, some aspects of the manuscript need to be revised before considering this work suitable for publication. Particularly the way in which the authors perform the stress treatments In the expression studies in yeast, that is, in my opinion, highly debatable. Also, qRT-PCR expression studies need statistical analysis.

My main concerns are related with the stress treatments performed in the heterologous expression of MaRab1 and MaRab7 genes in S cerevisiae:

a) Salt stress assay: in my first lecture I was very surprised to see growth of S cerevisiae strain after 3 days in the very extreme condition of 5 M NaCl in yeasts transformed with empty vector (pYES2). The same strain INVSc1 was used by Yu et al (2017) (https://doi.org/10.3390/ijms18112444) and these authors describe an absence of growth in 3 M NaCl, in fact these authors choose 2 M NaCl as a their limit of stringent salt stress condition  to use in screening experiments looking for stress-related genes. After checking the material and methods sections I see that the yeasts were resuspended in a 5 M NaCl solution and after that were plated on SC-Ura plates without any supplementation with salt, so the exposure to salt stress conditions was very brief…so in the additional 3 days of growth the yeasts probably have not been exposed to any stress…  This complementation assay should be repeated with SC-ura plates supplemented with NaCl (perhaps something like 1 M or even less) to maintain the stress condition during colony growth.

b) also, I am not very sure about the physiological meaning of an ABA external treatment in the yeast expression experiment, because S cerevisiae does not use ABA as a signal molecule, and because of that do not have ABA receptors or ABA signaling pathways…

Additional comments:

Regarding Abstract:

Line 30: I think that the authors’ work could give clues about the response of the olant to stress and not about the effects of the stress, because of that, I think that the sentence “into the molecular mechanism of salinity stress in M albus” should be changed with “molecular mechanism of the response of M. albus to saline stress”

Regarding Introduction:

Lines 44-45: please consider changing “In reaction to salinity stress, plants survive and maintain biomass in various ways” with “In response to saline stress, plants could develop several strategies to survive and maintain biomass”

Line 53: reference should be separated with a space from “well”

Lines 66-67: please change “plays a crucial role in multiple stress” with “plays a crucial role in multiple stress responses”

Lines 69-70: “have been induced to be expressed in reaction to cold, salt, drought, and ABA (Abscisic Acid) treatments” “have described as induced genes in response to cold, salt, drought, and Abscisic Acid (ABA) treatments”

Lines 74-74: the sentence is somewhat confusing accurate:  the RabA1 proteins do not “withstand against stress”. These proteins are part of a response of the plant that the plant develop to withstand against the deleterious effects of the stress, that in the particular case of these family of proteins “by mediating the transport of substances between the plasma membrane and the trans-Golgi.”. Please reformulate the sentence

Line 81-82: the sentence is cryptic…. which active role has PtRabE1b in the salt stress response of Populus trichocarpa?

Regarding results:

Lines 91-92: the sentence “Totals of 27, 30, 27 91 and 28 Rab genes were discovered in M. truncatula, G. max, C. arietinum and L. corniculatus, respectively.” should appear in the discussion section.

Line 94:  please change “Proteins with molecular weights 202 aa (22.43 kDa) to 235 aa (25.87 kDa) were produced by the MaRab gene” with ““MaRab genes coded for proteins with molecular weights 202 aa (22.43 kDa) to 235 aa (25.87 kDa)” (genes are segments of DNA that contain information, these structures do not produce anything)

Lines 95-99: the authors have not demonstrated experimentally the subcellular localization of the proteins described… so this sentence should be more accurate, reflecting the predictions of their putative cellular localizations according to the presence of signal domains in their amino acid sequences

Lines 110-112: this paragraph needs to be reformulated, since it contains a redundant description of the characteristics of the groups. Also, the sentence “During plant development, Group A seems to have expanded more than the other groups.” should stay in the discussion section

Lines 115-116: I think that the description of the results of figure 2 in the sentence “Arabidopsis had the most Group A members, with a total of 25. M. albus, O. sativa, M. truncatula, G. max, C. arietinum and L. corniculatus had 12, 14, 10, 14, 12 and 12 Group-A members, respectively.”, could be described in a more systematic and ordered way like “Arabidopsis was the species with the highest number of members (25), followed by O. sativa and G. max (14 members each), while M. albus, C. arietinum and L. corniculatus had 12 members, being  M. truncatula the species with the lower number (10) of group A members”.

Line 124: Resolution of the figure 2 should be improved (gene names are illegible). Also, the data of the phylogenetic tree, like gene names, distances…should appear as supplementary information

Lines 125-156: “Rab” and names of the species should appear in italics. Figure legend should describe color meaning

Line 128:  MEME software suite should be referenced

Line 147: I am not sure about the meaning of “the top”

Line 151-152: “In total, 401 cis-acting elements were found in this analysis and could be grouped into 24 sets according to their described roles in the bibliography”

Line 153: “The three sets with the higher numbers of elements were associated with…”

Line 154: MeJA abbreviation should be defined

Line 155: please change “As a result,” with “By the other side,”

Lines 156-160: the fact that each promoter has at least three cis-acting elements do not explain by itself the involvement of their genes in “numerous abiotic stress” responses or in hormone regulation…are these cis-acting elements stress-related or hormone-related? Please rewrite the sentence.

Line 165, 172, 174: Please use ABA abbreviation (since it was defined in line 70)

Line 167: “Group H could play”, since the authors have not demonstrated this fact experimentally

Line 178: resolution of figure 4 should be improved. The description of cis-elements is difficult to read

Line 214: resolution of figure 6 should be improved

Line 249: resolution of figure 7 should be improved. Smallest letters in panel B are illegible

Line 257: qRT-PCR results in Figure 8 need to include statistical analyses to check significant differences in gene expression levels

Line 262: “genes” are not elevated, their expression could be high or low…please rephrase

Line 263: Again, genes are DNA segments that do not perform tasks apart from containing information, the proteins coded by these genes are the biological structures that perform tasks in the cell response. Please rephrase.

Regarding discussion

Line 282: replace “…been published. [29,30].” With “…been published [29,30].”

Lines 335-338: Like as I have pointed out previously to see any growth of yeast transformed with empty vector pYES2 in 5 M salt. Also, the ABA treatment has no physiological sense because of the absence of ABA receptors or ABA transduction pathways in yeast.

Regarding Materials and Methods

Lines 372,378: please replace “e-value cutoff of 1e-5” with “e-value cutoff of 1x10-5”

Line 381: The software used to obtain the phylogenetic tree of plant Rab proteins should be detailed and referenced

Line 402: e-value < 1 x 10-10

Line 420: qRT-PCR data needs statistical analyses that should also be detailed.

Lines 410-411: I am not very sure abour the source of each RNAseq data used to obtain the transcriptome data used in the heatmap. It was used from previously available data or new transcriptomes were generated in this study?

If any RNAseq data was generated in this study the conditions for this obtention (library generation, sequencing methodology…) should be detailed, and the sequences uploaded in a database (it is the case of the tissues transcriptomes?)

If previously available RNAseq datasets were used, the they should be detailed and/or referenced (currently, this information is absent in the case of transcriptomes from stress treatments) .

Also, I think that a future reader could need a supplementary table with the accession IDs of the transcripts corresponding to each to the 27 MaRab genes in these transcriptome datasets.

Lines 435-436: I am not sure about how the camera “was used to o record the expression of binding proteins.”. As I have interpreted pictures were taken (with a camera) to check colony formation and growth….and no more than that.

Reviewer 2 Report

The authors have to cover the following concerns

1- Figures 4,6, and 6 are in poor quality and some details in the figures are not readable

2- Figure 7, please consider font size adjustment for titles and gene names in the heat map (I would decrease font size of gene names and increase the font size of the time points and conditions on the top)

3- Figure 8, you have to show statistical analysis and explain in materials and methods

4- During yeast expression, have you analyzed somehow the level of expression for genes of interest

5- I recommend citing these references in the introduction, methods, and discussion

a- Kamal, K. Y., Khodaeiaminjan, M., Yahya, G., El-Tantawy, A. A., Abdel El-Moneim, D., El-Esawi, M. A., Abd-Elaziz, M., & Nassrallah, A. A. (2021). Modulation of cell cycle progression and chromatin dynamic as tolerance mechanisms to salinity and drought stress in maize. Physiologia plantarum, 172(2), 684–695. https://doi.org/10.1111/ppl.13260

b- Yahya, G., Pérez, A. P., Mendoza, M. B., Parisi, E., Moreno, D. F., Artés, M. H., Gallego, C., & Aldea, M. (2021). Stress granules display bistable dynamics modulated by Cdk. The Journal of cell biology, 220(3), e202005102. https://doi.org/10.1083/jcb.202005102

c- Georgieva, M. V., Yahya, G., Codó, L., Ortiz, R., Teixidó, L., Claros, J., Jara, R., Jara, M., Iborra, A., Gelpí, J. L., Gallego, C., Orozco, M., & Aldea, M. (2015). Inntags: small self-structured epitopes for innocuous protein tagging. Nature methods, 12(10), 955–958. https://doi.org/10.1038/nmeth.3556

d- Moreno, D. F., Parisi, E., Yahya, G., Vaggi, F., Csikász-Nagy, A., & Aldea, M. (2019). Competition in the chaperone-client network subordinates cell-cycle entry to growth and stress. Life science alliance, 2(2), e201800277. https://doi.org/10.26508/lsa.201800277

e- Yahya, G., Hashem Mohamed, N., Pijuan, J., Seleem, N. M., Mosbah, R., Hess, S., Abdelmoaty, A. A., Almeer, R., Abdel-Daim, M. M., Shulaywih Alshaman, H., Juraiby, I., Metwally, K., & Storchova, Z. (2021). Profiling the physiological pitfalls of anti-hepatitis C direct-acting agents in budding yeast. Microbial biotechnology, 14(5), 2199–2213. https://doi.org/10.1111/1751-7915.13904

f- Amponsah, P. S., Yahya, G., Zimmermann, J., Mai, M., Mergel, S., Mühlhaus, T., Storchova, Z., & Morgan, B. (2021). Peroxiredoxins couple metabolism and cell division in an ultradian cycle. Nature chemical biology, 17(4), 477–484. https://doi.org/10.1038/s41589-020-00728-9

g- Yahya, G., Wu, Y., Peplowska, K., Röhrl, J., Soh, Y. M., Bürmann, F., Gruber, S., & Storchova, Z. (2020). Phospho-regulation of the Shugoshin - Condensin interaction at the centromere in budding yeast. PLoS genetics, 16(8), e1008569. https://doi.org/10.1371/journal.pgen.1008569

Round 2

Reviewer 1 Report

I am happy to see that the authors have addressed most of the issues pointed out before, and that the revised version of the manuscript has improved with the changes made. Still, there are a few issues that should be addressed before considering the acceptance of the work.

Regarding my concerns about the yeast complementation experiments:

Looking the references pointed out by the authors in their reply to my comments, I see that my initial concern was due basically about a lack of information. Essentially you are repeating the experiment performed previously in your laboratory (From Zhang et al , 2020 (https://doi.org/10.1002/csc2.20362), where  the saline stress treatment consists in the  resuspension of yeasts in 5 M NaCl, but leaving them in this condition for 36 hours and after that the cells are leave to grown in solid SC-Ura media.

This particular detail should be included in section 4.6 of material and methods and also in figure 9 legend, to avoid that a future reader could interpret that yeasts have been subjected to a brief episode of stress (i.e cells resuspended and immediately spotted into SC-Ura media, that was my initial interpretation with the actual description of the experiment).

Regarding Introduction

Line 74: “have been described as induced genes” (sorry, my mistake with my previous suggestion)

Line 80: “RabA1 proteins” (without italics, I must insist: proteins do the molecular tasks in the cell and not their genes)

Regarding Results

Lines 236-237: figure 6 appear duplicated. Please delete the version with low resolution

Line 275: Figure 7 appears also duplicated

Line 306: Figure 9 appears also duplicated

Reviewer 2 Report

I am pleased to accept the manuscript in this form, congratulations.

Author Response

We are pleased to hear that. The authors are grateful for the valuable and helpful comments of the reviewers.